# IIAEK Targets Intestinal Alkaline Phosphatase (IAP) to Improve Cholesterol Metabolism with a Specific Activation of IAP and Downregulation of ABCA1

**DOI:** 10.3390/nu12092859

**Published:** 2020-09-18

**Authors:** Asahi Takeuchi, Kentaro Hisamatsu, Natsuki Okumura, Yuki Sugimitsu, Emiko Yanase, Yoshihito Ueno, Satoshi Nagaoka

**Affiliations:** 1Laboratory of Molecular Function of Food, Department of Applied Life Science, Faculty of Applied Biological Sciences, Gifu University, 1-1 Yanagido, Gifu 501-1193, Japan; z4521051@edu.gifu-u.ac.jp (A.T.); 8polonium4@gmail.com (K.H.); natsunatsu11294@gmail.com (N.O.); nome.imakev.og@gmail.com (Y.S.); 2Laboratory of Bio-organic Chemistry, Department of Applied Life Science, Faculty of Applied Biological Sciences, Gifu University, 1-1 Yanagido, Gifu 501-1193, Japan; e-yanase@gifu-u.ac.jp (E.Y.); uenoy@gifu-u.ac.jp (Y.U.)

**Keywords:** IIAEK, alkaline phosphatase, photoaffinity labeling, Caco-2 cells, cholesterol, ABCA1

## Abstract

IIAEK (Ile-Ile-Ala-Glu-Lys, lactostatin) is a novel cholesterol-lowering pentapeptide derived from bovine milk β-lactoglobulin. However, the molecular mechanisms underlying the IIAEK-mediated suppression of intestinal cholesterol absorption are unknown. Therefore, we evaluated the effects of IIAEK on intestinal cholesterol metabolism in a human intestinal model using Caco-2 cells. We found that IIAEK significantly reduced the expression of intestinal cholesterol metabolism-associated genes, particularly that of the ATP-binding cassette transporter A1 (ABCA1). Subsequently, we chemically synthesized a novel molecular probe, IIXEK, which can visualize a complex of target proteins interacting with photoaffinity-labeled IIAEK by fluorescent substances. Through photoaffinity labeling and MS analysis with IIXEK for the rat small intestinal mucosa and intestinal lipid raft fractions of Caco-2 cells, we identified intestinal alkaline phosphatase (IAP) as a specific molecule interacting with IIAEK and discovered the common IIAEK-binding amino acid sequence, GFYLFVEGGR. IIAEK significantly increased IAP mRNA and protein levels while decreasing ABCA1 mRNA and protein levels in Caco-2 cells. In conclusion, we found that IIAEK targets IAP to improve cholesterol metabolism via a novel signaling pathway involving the specific activation of IAP and downregulation of intestinal ABCA1.

## 1. Introduction

Lifestyle diseases, especially arteriosclerosis-related disorders, are closely associated with diet. Improving cholesterol metabolism through diet modifications is an important strategy for the prevention and treatment of hypercholesterolemia and hyperlipidemia [1]. Dietary proteins and peptides are effective in ameliorating dyslipidemia and hypercholesterolemia in animals and humans [2]. However, the molecular mechanisms underlying their cholesterol-lowering effects have not yet been elucidated. We previously discovered a cholesterol metabolism-improving pentapeptide: IIAEK (lactostatin). It was obtained from β-lactoglobulin trypsin hydrolysate (LTH), which is a major component of the milk whey protein [3]. We observed that the serum cholesterol-lowering effect of IIAEK (Ile-Ile-Ala-Glu-Lys, lactostatin) was comparable to that of beta-sitosterol, which is a phytosterol that was described as having pharmaceutical potential based on a study in rats [3]. Since cholesterol is converted to bile acids and excreted in the liver, the effect of IIAEK on cholesterol 7alpha-hydroxylase (CYP7A1), a rate-limiting enzyme in bile acid synthesis, was evaluated. The modulation of CYP7A1 gene expression, which is crucial for the prevention and improvement of hyperlipidemia and atherosclerosis, has been reported to improve these conditions in animal models overexpressing CYP7A1 [4,5]. The addition of IIAEK to HepG2 cells, which are human hepatocarcinoma-derived cells, significantly increased their CYP7A1 mRNA levels relative to the control [6]. In addition, the IIAEK-induced upregulation of CYP7A1 mRNA was triggered through the calcium channel-mediated mitogen-activated protein kinase (MAPK) signaling pathway [6]. We proposed that IIAEK might improve cholesterol metabolism via a specific membrane protein (e.g., membrane receptor) recognized by IIAEK in HepG2 cells [6]. Interestingly, IIAEK is not degraded by digestive enzymes such as pepsin and trypsin. Therefore, it reaches the small intestine prior to intestinal absorption. However, the mechanism through which IIAEK directly affects small intestinal epithelium is unclear. Our hypothesis is that the target protein (such as a receptor) of IIAEK is associated with a novel regulatory pathway responsible for intestinal cholesterol metabolism. Currently, there is no information about food-derived bioactive peptide-related specific receptors, including IIAEK.

Therefore, in the present study, photoaffinity labeling was performed to visualize target proteins interacting with IIAEK, and fluorescently labeled probes were chemically synthesized to capture the target proteins. Photoaffinity labeling, a technique developed by Westheimer in 1962, utilizes the reversible interaction of a bioactive compound with a target protein [7]. Generally, protein–ligand interactions are characterized by transient, non-covalent interactions. However, in photoaffinity labeling, transient protein–ligand interactions are analyzed by the formation of covalent complexes with photoreactive groups, thereby enabling the identification of interacting amino acid sequences [8]. In addition, we utilized the click reaction to visualize the photoaffinity-labeled target protein–IIAEK complex. The click reaction, first described by Sharpless et al., is a covalent bond-forming reaction that functions effectively during molecular modifications and coupling reactions [9]. The most well-known click reaction is the cycloaddition of azides and alkynes (Huisgen cycloaddition) reported by Sharpless et al. [10] and Meldal et al. [11]. In photoaffinity labeling, the ligand–target protein complex is irreversibly covalently cross-linked by UV irradiation. Then, this complex is labeled with a fluorescent azide (rhodamine) using the click reaction, and finally, the fluorescent azide-labeled captured target protein complex is separated and identified using SDS-PAGE and MS, respectively. In the present study, we used a human small intestinal epithelial in vitro model (Caco-2 cells) to identify intestinal cholesterol absorption-associated genes affected by IIAEK, with a particular focus on ABCA1. In addition, we performed photoaffinity labeling and MS analysis with the chemically synthesized novel molecular probe IIXEK in rat small intestinal mucosal fractions and intestinal lipid raft fractions from Caco-2 cells in order to identify the target molecules interacting with IIAEK in the small intestinal epithelium. Furthermore, we assessed the relationship between intestinal ABCA1 and IAP (intestinal alkaline phosphatase), the specific molecule shown to interact with IIAEK in Caco-2 cells.

## 2. Materials and Methods

### 2.1. Cell Culture

Caco-2 cells were acquired from the American Type Culture Collection (Manassas, VA, USA). The cells were maintained in Dulbecco’s modified Eagle’s medium (DMEM) supplemented with 10% fetal calf serum, 4 mmol/L l-glutamine, 50,000 IU/L penicillin, and 50 mg/L streptomycin. The cells were either seeded in 6-well Transwell^®^ plates (Corning, Inc., Corning, NY, USA) at a density of 2 × 10^5^ cells/well or in 75 mm Transwell^®^ plates (Corning, Inc., Corning, NY, USA) at a density of 18.8 × 10^5^ cells/well, and they were grown for 14 days post-confluence.

### 2.2. Chemicals

IIAEK [Ile-Ile-Ala-Glu-Lys: purity (>95%)] and IIXEK (purity > 95%) were obtained from the Peptide Institute, Inc. (Osaka, Japan). Cholesterol and sodium taurocholate were purchased from Sigma Aldrich (St. Louis, MO, USA).

### 2.3. Cholesterol Absorption Assay for Caco-2 Cells

Caco-2 cells were grown in 6-well Transwell^®^ plates (Corning, Inc., Corning, NY, USA) containing 0.5 mL DMEM supplemented with fetal bovine serum for 14 days post-confluence. The absorption of [^14^C]-labeled micellar cholesterol in Caco-2 cells was measured as previously described [12,13], with some modifications. Briefly, on day 14, [^14^C]-labeled micellar solutions containing 3.7 kBq [4-^14^C]-cholesterol (2.0 Gbq/mmol, NEN) with or without 2 mM IIAEK, 5 mM taurocholate, and 250 μM cholesterol in serum-free DMEM were prepared by sonication and incubated with shaking at 37 °C for 24 h. Following incubation, cholesterol absorption was measured for 24 h. The cellular protein concentration was determined using a commercially available kit (Bio-Rad Protein Assay; Bio-Rad, Hercules, CA, USA).

### 2.4. RNA Isalation from Caco-2 Cells and Real-Time PCR Using TaqMan Probe

Caco-2 cells were treated with or without 2 mM IIAEK for 24 h. Micelles consisted of 5 mM taurocholate and 250 μM cholesterol as previously described [13]. Following treatment, total RNA was isolated from Caco-2 cells using a NucleoSpin^®^ RNA Kit (MACHEREY-NAGEL, Düren, Germany) and was treated with DNase I (RNase-Free DNase; Qiagen, 79254, Hilden, Germany). Total RNA was reverse transcribed into cDNA using a High-Capacity cDNA Archive Kit (Applied Biosystems, Foster City, CA, USA). Real-time PCR was performed on an ABI PRISM 7000, using the TaqMan^®^ Universal PCR Master Mix (Applied Biosystems), according to the manufacturer’s protocol, as previously described [14]. TaqMan^®^ Ribosomal RNA Control Reagents (Applied Biosystems) were used as primers, and a TaqMan^®^ probe was used for the 18S ribosomal RNA gene-based PCR. The primers and TaqMan^®^ probes for human scavenger receptor B-1 (SR-B1, Hs00969821_m1), Niemann-Pick C1-like 1 (NPC1L1, Hs00203602_m1), ATP-binding cassette transporter A1 (ABCA1, Hs00194045_m1), acetyl-CoA acetyltransferase 2 (ACAT2, Hs00255067_m1), microsomal triglyceride transfer protein (MTP, Hs00165177_m1), ATP-binding cassette transporter G5 (ABCG5, Hs03037375_m1), ATP-binding cassette transporter G8 (ABCG8, Hs00223690_m1), cytochrome P450, family 27, subfamily A, polypeptide 1 (CYP27A1, Hs01026016_m1), and 18S ribosomal RNA (4308329) were purchased from Applied Biosystems, as part of a TaqMan^®^ Gene Expression Assay. The following primers were used with the StepOnePlus^TM^ real-time PCR system (Applied Biosystems) and the SYBR^®^ Premix Ex Taq (TAKARA, Tokyo, Japan): for human SR-B1, TCCTCACTTCCTCAACGCTG (forward) and TCCCAGTTTGTCCAATGCC (reverse); for NPC1L1, ATCTTAACTGTCGGATCCACAAAAA (forward) and AACCTGATGGCATTGTGAGACAT (reverse); for ABCA1, TGCTGCATAGTCTTGGGACTC (forward) and ACCTCCTGTCGCATGTCACT (reverse); for ACAT2, CCGGAAGATGTGTCTGAGGT (forward) and CACCCACACTGGCTTGTCTA (reverse); for MTP, ACCTGCAGACGTGTATTCATTC (forward) and CCCAGCTAGGAGTCACTGAGA (reverse); for ABCG5, CCGACTGATTGGCAACTACA (forward) and GCTCATCAAACAGCATGACC (reverse); for ABCG8, AACTTGAGCAGCCTGTGGA (forward) and CATCAGCCCTTCAAAACACC (reverse); for CYP27A1, GTGCTGCCTTTCTGGAAGCGAT (forward) and TAGCCAGACACCTGGATGCCAT (reverse); and for 18S ribosomal RNA, CTCAACACGGGAAACCTCAC (forward) and CGCTCCACCAACTAAGAACG (reverse).

### 2.5. RNA Isolation from Caco-2 Cells and Real-Time PCR

Caco-2 cell treatment, and RNA and cDNA preparations were performed as previously described [12]. Real-time PCR was performed using the StepOnePlus^TM^ Real-time PCR system (Applied Biosystems) and SYBR^®^ Premix Ex Taq (TAKARA), according to the manufacturer’s protocol. The following primers were used in the assay: for ABCA1, TGCTGCATAGTCTTGGGACTC (forward) and ACCTCCTGTCGCATGTCACT (reverse); for intestinal alkaline phosphatase (IAP), CATACCTGGCTCTGTCCAAGA (forward) and GTCTGGAAGTTGGCCTTGAC (reverse); for Liver X receptor α (LXRα), TGGACACCTACATGCGTCGCAA (forward) and CAAGGATGTGGCATGAGCCTGT (reverse); for Liver X receptor β (LXRβ), CTTCGCTAAGCAAGTGCCTGGT (forward) and CACTCTGTCTCGTGGTTGTAGC (reverse); and for 18S ribosomal RNA, CTCAACACGGGAAACCTCAC (forward) and CGCTCCACCAACTAAGAACG (reverse).

### 2.6. Western Blot Analysis of Caco-2 Cells

Caco-2 cells were treated with or without 2 mM IIAEK for 48 h for ABCA1 protein level with cholesterol micelle or 24 h for IAP and ABCA1 protein levels. The micelle was prepared as previously described [13]. Protein preparation from Caco-2 cells and Western blot analyses were performed as previously described [12]. For Western blot analysis, we used the following specific antibodies: anti-ABCA1 (sc-58219, Santa Cruz Biotechnology, Dallas, TX, USA), anti-IAP (ab186422, Abcam, Cambridge, UK), and anti-β-actin (sc-47778, Santa Cruz). Western blot analysis was performed using an ImmunoStar^®^ LD system (Wako Pure Chemical Industries, Osaka, Japan).

### 2.7. Transient Transfections and Luciferase Assay Using Caco-2 Cells

Protein preparation from Caco-2 cells and luciferase analyses were performed as previously described [12]. Following plasmid (ABCA1-Luc plasmid containing −928 to +107 bp, −536 to +107 bp, −126 to +107 bp, or −36 to +107 bp of the ABCA1 gene promoter, and pPGK β-galactosidase plasmid) transfection, the cells were treated with 2 mM IIAEK for 12 h with micelles, which were prepared as previously described [13]. Subsequently, the cells were lysed with reporter lysis buffer (Promega, Madison, WI, USA). Luciferase activity was measured using a luciferase assay system (Promega) on a Fluoroskan Ascent FL instrument (Labsystems) according to the manufacturer’s instructions. β-galactosidase activity was measured using a β-galactosidase ELISA Kit (Promega). Then, luciferase activity was normalized to that of β-galactosidase.

### 2.8. Alkaline Phosphatase Measurement

Alkaline phosphatase (AP) activity was measured as previously described [15,16] with some modifications. Caco-2 cells were treated with or without 2 mM IIAEK. Micelles were prepared as previously described [13]. Following treatment, the cells were washed twice with 0.9% NaCl. Subsequently, 691 μL of cold 50 mM Tris buffer [pH 7.5, Trizma base (Sigma-Aldrich, St. Louis, MO, USA, T1503)] was added. Protein was collected on ice and homogenized with an injection needle (26G × 1/2) (TERUMO, Tokyo, Japan, NN-2613S). AP activity was estimated using 1.25 mg/mL disodium p-nitrophenol phosphate as a substrate in Tris-HCl buffer [pH 10.0, 5 mM MgCl_2_._6_H_2_O (Wako, 135-00165) and 200 mM Trizma base (Sigma-Aldrich, St. Louis, MO, USA, T1503)]. After 30 min of incubation at 37 °C, the absorbance at 405 nm was measured, and AP activity was calculated as μmol/min using a calibration curve for various concentrations of p-nitrophenol. μmol/min was defined as U. U was converted to U/mg protein in order to normalize AP activity to the protein concentration, which was determined using a commercially available kit (BioRad, protein assay; BioRad).

### 2.9. Preparation of Rat Intestinal Mucosal Protein

Intestinal mucosal proteins was obtained from 5-week-old male Wistar rats (Japan SLC, Hamamatsu, Japan). Following a 12 h fast, the rats were anesthetized, and the upper part of the small intestine was removed, cut open, and washed with cold saline. Subsequently, the small intestinal mucosal tissue was collected, and the membrane proteins were extracted using Extraction Buffer 2B of the ProteoExtract^®^ Transmembrane Protein Extraction Kit (Novagen, Merck KGaA, Darmstadt, Germany, 71772) according to the manufacturer’s protocol. The Ethics Committee on Animal Experiments at Gifu University approved all the experimental protocols performed (permit number: 15124). All experiments during this study were performed in accordance to the experimental guidelines and regulations of Gifu University.

### 2.10. Preparation of Intestinal Lipid Rafts from Caco-2 Cells

Caco-2 cells were treated with or without 2 mM IIAEK for 24 h. Following treatment, the cells were washed twice with ice-cold phosphate-buffered saline and collected in 1 mL of cold TNE buffer (25 mM Tris-HCl, 150 mM NaCl, 1 mM EGTA, pH 7.5) + 1% (*w*/*v*) Triton X-100. The cell lysates were homogenized on ice using a glass homogenizer (WHEATON, 903475) with an attached potter-type shuttle (WHEATON, 358034). The homogenate was transferred to the 13PA tube (HITACHI: 332001A), and 2 mL of 80% (*w*/*v*) sucrose in TNE buffer was added. Homogenates were layered with 6 mL of sucrose density gradient solution (5–30%) using a fractionator, followed by ultracentrifugation at 200,000× *g* and 4 °C for 17 h in an ultracentrifuge (Himac, CP80WX, HITACHI, Tokyo, Japan) with a swing rotor (Himac, P40ST, HITACHI), as previously described [17]. Subsequently, samples were fractionated into 10 fractions and recovered using the density gradient fractionator MODEL DGF-U (HITACHI). Sucrose density gradient formation was confirmed by measuring the sucrose density (*w*/*w* %) of each fraction with the ATAGO pocket sugar meter (ATAGO). Each fraction was subjected to ultrafiltration using an Amicon^®^ Ultra 10K device (Millipore) according to the manufacturer’s protocol. The obtained concentrated protein solutions obtained were used for photoaffinity labeling.

### 2.11. Photoaffinity Labeling and Nano LC-MS/MS Analyses

DMSO (0.5 μL) was added to rat intestinal mucosal protein fractions or intestinal lipid rafts from Caco-2 cells at a rate of 1 mg/mL, followed by incubation on ice for 15 min. Subsequently, 10 mM IIXEK (0.5 μL) dissolved in DMSO was added, again followed by incubation on ice for 15 min. Photoaffinity labeling was performed as previously described [18]. Then, the protein fractions and lipid rafts were subjected to UV irradiation at 365 nm for 30 min, and 302 nm for 10 min. Later, 2.5 μL of 10% SDS and 0.5 μL of 5 mM Azide-fluor488 (Sigma-Aldrich) were added. Next, 2 μL of the catalytic mixture [1.5 μL of 1.7 mM TBAT (TCI), 0.5 μL of 50 mM CuSO4 (Sigma-Aldrich), and 0.5 μL of 50 mM Sodium l-ascorbate (Sigma-Aldrich)] were added, followed by incubation at 32 °C for 30 min. Subsequently, this mixture was subjected to 4–12% SDS-PAGE for fluorescence scanning. The BenchMark™ Fluorescent Protein Standard was used as the molecular weight marker. The photoaffinity labeled protein was analyzed by nano LC-MS/MS (Q Exactive Plus, Thermo Scientific, Waltham, MA, USA). A Mascot search engine (www.matrixscience.com) was used for the database search within UniprotKB/Swiss-Prot database to match with the parent proteins.

### 2.12. Statistical Analysis

Values are expressed as means ± SEM. The statistical significance of the differences between two groups was evaluated by Student’s *t*-tests [19]. The differences were considered significant when * *p* < 0.05, ** *p* < 0.01, *** *p* < 0.001.

## 3. Results

### 3.1. Effect of IIAEK on Cholesterol Absorption in Caco-2 Cells

First, we measured the effect of IIAEK on intestinal cholesterol absorption in Caco-2 cells. Treatment of Caco-2 cells with 2 mM IIAEK resulted in a significant reduction in intestinal cholesterol absorption (Figure 1A).

### 3.2. Effect of IIAEK on the mRNA Levels of Cholesterol Metabolism-Associated Genes in Caco-2 Cells

To explore IIAEK’s effects on cholesterol absorption via IIAEK at the gene expression level, we investigated whether IIAEK affected the expression of intestinal cholesterol metabolism-associated genes such as SR-B1, NPC1L1, ABCA1, ACAT2, MTP, ABCG5, ABCG8, CYP27A1, LXRα, and LXRβ in differentiated Caco-2 cells. The cells were treated with 2 mM IIAEK or vehicle for 24 h. Compared to the controls, 2 mM IIAEK treatment results in significantly lower mRNA levels of SR-B1, ABCA1, ACAT2, MTP, CYP27A1, and LXRα in differentiated Caco-2 cells cultured with cholesterol micelles (Figure 1B). A tendency of lower LXRβ expression was also observed in Caco-2 cells treated with 2 mM IIAEK (Figure 1B).

### 3.3. Effect of IIAEK on ABCA1 Protein Levels in Caco-2 Cells

In order to assess whether IIAEK affects intestinal ABCA1 levels, we measured ABCA1 protein expression in Caco-2 cells via Western blot analysis. Compared to controls, 2 mM IIAEK treatment significantly reduced ABCA1 protein levels in Caco-2 cells (Figure 1C,D).

### 3.4. Effect of IIAEK on ABCA1 Gene Promoter Activity in Caco-2 Cells

We investigated the effect of IIAEK on human ABCA1 gene promoter activity in Caco-2 cells. We also employed three ABCA1 gene promoter deletion mutants (ABC–536, ABC–126, and ABC–36) in order to identify the specific region implicated in the IIAEK-mediated downregulation of ABCA1 expression. Caco-2 cells were treated with or without 2 mM IIAEK for 12 h, and the cell lysate was subjected to a luciferase assay. We observed that 2 mM IIAEK treatment significantly reduced ABCA1 gene promoter activity. Further, deletion of the LXR and SP1 response region (ABC–36) within the ABCA1 promoter prevented the IIAEK-induced decrease in ABCA1 gene promoter activity (Figure 1E).

### 3.5. Chemical Synthesis of IIXEK, a Molecular Probe for Photoaffinity Labeling

We used photoaffinity labeling to capture target proteins interacting with IIAEK. Photoaffinity labeling involves the reversible interaction of a bioactive compound (e.g., IIAEK) with its target protein. The bioactive compound and target protein are mixed, irradiated with light, and a molecular probe with a photoreactive group is used to cross-link the bioactive compound and target protein with an irreversible covalent bond, thereby labeling them with fluorescent molecules. The click reaction was used to detect the target protein–IIAEK complex, following photoaffinity labeling using photoreactive groups. As previously described [18,20], we combined a diazirine moiety for target protein capture and a terminal alkyne for detecting the target protein complex to synthesize 3-ethynyl-5-[3-(trifluoromethyl)-3H-diazirin-3-yl] benzoic acid. Finally, we evaluated a site for introducing the chemically synthesized molecular probe into IIAEK. In our previous studies, we assessed the effects of IIAEK, its fragment peptides, and constituent amino acids on CYP7A1 mRNA levels and found that the EK site was important for the biological effect of IIAEK [6]. Therefore, we retained the EK site and introduced the probe into the closest alanine residue located at the EK site. However, since the alanine residue lacked an amino group, it was replaced with (S)-2,3-diaminopropionic acid. The probe was introduced into the side chain amino group of (S)-2,3-diaminopropionic acid by the formation of an amide bond (Figure 2A). The synthesized probe was consigned to Peptide Institute, Inc. (Osaka, Japan). Thus, IIXEK, a mutant form of IIAEK possessing 3-ethynyl-5-[3-(trifluoromethyl)-3H-diazirin-3-yl] benzoic acid, was synthesized by a 9-fluorenylmethyloxycarbonyl (Fmoc) strategy (Figure 2B). Upon UV irradiation, this IIXEK forms covalent bonds with the target proteins (Figure 2B). Furthermore, to confirm the occurrence of the click reaction, rhodamine was introduced into IIXEK with diazirine and terminal alkyne in the presence of a copper ion catalyst. Subsequently, we tested for the occurrence of the Huisgen cycloaddition reaction (Figure 2C). The click reaction began at room temperature. At 1 h after reaction initiation, the analysis was performed using LC-MS (JEOL, JMS-T100TD, Instrumental Analysis Field, Gifu University Life Science Research Support Center). The peak generated due to the click reaction was detected at *m*/*z* 1397.82 [M − H]^−^ (Figure 2D).

### 3.6. Photoaffinity Labeling and MS Analysis of the Intestinal Lipid Raft Fractions Derived from Caco-2 Cells and Rat Intestinal Mucosal Protein by a Novel Molecular Probe, IIXEK

We demonstrated that IIAEK decreased intestinal cholesterol absorption in Caco-2 cells by downregulating the expression of intestinal cholesterol metabolism-associated genes (Figure 1A,B). Based on these results, we hypothesized that IIAEK might affect the expression of intestinal cholesterol metabolism-associated genes through intracellular signaling. Therefore, we focused on lipid rafts as signaling platforms and developed an experimental method to extract the lipid rafts of Caco-2 cells via a sucrose density gradient. The level of flotillin-1, a lipid raft marker protein, was measured. Results revealed that it accumulated in fractions (Fr.) 2 to 4 (Figure 3A,B). We also measured AP activity, another lipid raft marker, and we found a peak in AP activity in Fr. 3 and 4 (Figure 3C). Compared to the control group, treatment with 2 mM IIAEK resulted in an approximately 3-fold increase in AP activity (Figure 3C). We performed photoaffinity labeling and nano LC-MS/MS analysis of IIXEK in Fr. 3 and found that human IAP interacted with IIAEK, with the complex being observed at the 87.4 kDa band (Figure 3D). In addition, we found that IAP-2, represented by a fluorescent band at 94.4 kDa, is a target protein interacting with IIAEK (Figure 3E). We concluded that the 87.4 kDa band represents human IAP revealed by nano LC-MS/MS data (sequences coverage, 14.77%, Figure 4A). Matched peptide sequences are bold-faced and underlined. In addition, we concluded that the 94.4 kDa band was identified as rat IAP-2 revealed by nano LC-MS/MS data (sequences coverage, 37.02%, Figure 4D). Matched peptide sequences are bold-faced and underlined. Interestingly, comparison of the IIXEK-binding amino acid sequences of IAP in the intestinal lipid raft fraction of Caco-2 cells (IAP) (Figure 4A) and rat small intestinal mucosal protein fractions (IAP-2) (Figure 4D) by MS analysis revealed that GFYLFVEGGR was the common IIAEK-binding amino acid sequence, and their position within human IAP (Uniprot accession No. P09923) from N to C terminal: 324–333 and rat IAP-2 (Uniprot accession No. P51740) from 324 to 333 (Figure 4A,D). In the lipid raft fraction from Caco-2 cells, we identified the 57.8 kDa band as ATP synthase subunit beta, mitochondrial (coverage, 74.67%, Uniprot accession No. P06576, Figure 4B) and 52.6 kDa band as trifunctional enzyme subunit beta, mitochondrial (coverage, 71.52%, Uniprot accession No. P55084, Figure 4C) from nano LC-MS/MS data sequences. Matched peptide sequences are bold-faced and underlined. In rat mucosal protein, we identified the 56.3 kDa band as ATP synthase subunit beta, mitochondrial (coverage, 74.67%, Uniprot accession No. P10719, Figure 4E) from nano LC-MS/MS data sequences. Matched peptide sequences are bold-faced and underlined.

### 3.7. IIAEK Induces IAP Activation and Down-Regulation of ABCA1 in Caco-2 Cells

Caco-2 cells were treated with 2 mM IIAEK or vehicle for 24 h. Compared to the controls, IIAEK significantly increased AP activity in Caco-2 cells incubated with or without cholesterol micelles (Figure 5A,B). Moreover, IIAEK significantly increased IAP mRNA levels in cells incubated with or without cholesterol micelles, compared to the controls (Figure 5C,D). We also investigated the expression of cholesterol metabolism-associated gene ABCA1. IIAEK significantly reduced ABCA1 mRNA levels in Caco-2 cells incubated with or without cholesterol micelles (Figure 5E,F). In addition, IIAEK significantly increased IAP protein levels in Caco-2 cells incubated without cholesterol micelles (Figure 5G,H) and decreased those of ABCA1 (Figure 5I,J).

## 4. Discussion

Herein, we observed that the cholesterol-lowering pentapeptide IIAEK affects various intestinal cholesterol metabolism-related genes (SR-B1, ABCA1, ACAT2, and MTP) and has an inhibitory effect on cholesterol absorption in Caco-2 cells. In particular, IIAEK treatment significantly decreased the protein levels of ABCA1 as well as gene promoter activity (Figure 1A–E). Based on Figure 1B–D, as 27-OH cholesterol is a ligand for LXR, we presume that the significant IIAEK-induced decrease in the mRNA level of CYP27A1 may result in a decrease of 27-OH cholesterol levels, a decrease of LXRα mRNA expression, as well as decreased ABCA1 mRNA and protein levels [21]. Therefore, in order to identify the site within the ABCA1 promoter that is required for the downregulation of ABCA1 by IIAEK, we conducted luciferase assays with a number of ABCA1 promoter deletion mutants (Figure 1E). We found that deletion of the LXR and SP1 response element within the ABCA1 promoter prevented the IIAEK-induced decrease in ABCA1 promoter activity (Figure 1E). Taken together, Figure 1B,E suggests that the LXR response element may be required for the IIAEK-induced decrease in ABCA1 mRNA expression. Intestinal ABCA1 functions as a regulator of cholesterol absorption [22]. Interestingly, a recent study in mice reported that intestinal ABCA1 knockout reduced cholesterol absorption by ≈28% and total plasma cholesterol by approximately 30% [22]. Our previous study revealed that the suppression of intestinal cholesterol absorption by a novel cholesterol-lowering dipeptide, phenylalanine-proline (FP), was associated with a reduction in intestinal ABCA1 expression levels [12]. These results suggest that the inhibitory effect IIAEK exerts on cholesterol absorption is likely to be accompanied by a decrease in the expression of ABCA1 in Caco-2 cells.

There is no information on specific receptors for food-derived bioactive peptides, including IIAEK. For the first time, we identified target proteins interacting with IIAEK on the surface of small intestinal epithelial cells using a novel molecular probe, IIXEK. IIXEK was chemically synthesized and linked to fluorescent molecules in order to visualize target protein complex interacting with photoaffinity-labeled IIAEK (Figure 2A–D). Photoaffinity labeling and MS analysis of the small intestinal mucosal protein fractions of rats were performed using IIXEK. Lipid raft fractions extracted from Caco-2 cells were also subjected to photoaffinity labeling and MS analysis (Figure 3D). Human IAP was detected in the 87.4 kDa band. IAP is a brush border enzyme highly expressed in the small intestine [23]. Humans have at least four isotypes of AP: tissue-nonspecific AP (TNAP), which is expressed in the liver, bones, and kidney; intestinal AP (IAP), which is localized in the small intestine; placental AP (PAP); and germ cell AP (GCAP) [24]. AP, including IAP, is a glycoprotein, and is composed of about 20% carbohydrates [25]. While all APs are translated from the same mRNA transcript, depending on their location in the intestine, they may undergo various glycosylation modifications [26]. Previous studies have revealed that the molecular weight of rat IAP is 93 kDa [27,28].

The molecular weight of the novel molecular probe IIXEK used in this photoaffinity labeling experiment is 0.82387 kDa (value calculated from the structural formula of IIXEK). IIXEK forms irreversible covalent bonds with target proteins upon UV irradiation, and rhodamine (0.57458 kDa) was used to detect the target protein–IIXEK complex. We have discovered that there is a common IIAEK binding amino acid sequence (GFYLFVEGGR) of IAP in both Caco-2 cells and rat intestinal mucosa (Figure 4A,D). Therefore, it was observed that one IIXEK molecule (0.82387 kDa) bound to the 93 kDa rat IAP result in a rat IAP–IIXEK–rhodamine complex detected at around 94.39847 kDa by SDS-PAGE. Subsequently, we studied target molecules interacting with IIAEK in the lipid raft fraction extracted from Caco-2 cells. Previous reports have shown that in a 5% SDS-PAGE gel, the human IAP subunit can be detected at 86 kDa, and the human IAP dimer can be observed at 170 kDa [29]. Since human IAP contains one common IIXEK-binding sequence in each subunit, we suggested that a single IIXEK molecule (0.82387 kDa) binds to the 86 kDa human IAP subunit. In the current study, we detected the band of the human IAP–IIXEK–rhodamine complex at around 87.39847 kDa on the SDS-PAGE gel. In the rat small intestinal mucosal fractions, IAP-2 was detected at 94.4 kDa (Figure 3E). In Figure 2D, the click reaction product formation of IIXEK and rhodamine has been confirmed. Setting the control run without IIXEK in order to show that IIXEK binding shifts the molecular weight by 0.82387 kDa will be considered for future experiments. Of note, a number of bands were observed on the SDS-PAGE gel following photoaffinity labeling by IIXEK using lipid raft fractions from Caco-2 cells (Figure 3D) or rat intestinal mucosal protein (Figure 3E). At first, we concluded that the 87.4 kDa band represented human IAP identified from nano LC-MS/MS data (sequences coverage, 14.77%, Figure 4A). We also concluded that the 94.4 kDa band was identified as rat IAP revealed by nano LC-MS/MS data (sequences coverage, 37.02%, Figure 4D). Moreover, both the 87.4 kDa band and 94.4 kDa band contained the amino acid sequence of IAP, including the GFYLFVEGGR (common IIAEK-binding amino acid sequence) revealed by MS analysis (Figure 4A,D). Since the binding specificity of IIXEK for IAP during photoaffinity labeling is not particularly high, as revealed by SDS-PAGE (Figure 3D,E), we also investigated the relationships between IAP and IIAEK in the lipid raft fractions from Caco-2 cells. As it is well known that AP is a typical intestinal lipid raft marker as well as flotillin-1, we measured AP activity to confirm the formation of the lipid raft and found that AP activity was drastically increased by IIAEK in the lipid raft fractions, which is of great importance with regard to cellular signal transduction, including cholesterol metabolism (Figure 3C). Furthermore, IAP protein level was significantly increased by IIAEK treatment compared to control in Caco-2 cells (Figure 5G). Finally, IIAEK significantly increased both the IAP mRNA level and AP activity in Caco-2 cells. Taken together, we have concluded that an important interaction occurs between IAP and IIAEK. In order to show the specificity of the interaction between IIAEK and IAP, we need to use a control peptide in a future study and confirm that this peptide will not label IAP.

We compared the IIXEK-binding amino acid sequence of rat IAP-2 with human IAP (Figure 4A,D). Surprisingly, GFYLFVEGGR is shared between IAPs of the two samples (Figure 4A,D). This common amino acid sequence was also found in Akp3, which is one of the murine IAPs. The glutamic acid (E) in this sequence is regulated by Mg^2+^ in the rat IAP, along with aspartic acid (D), the 42nd amino acid residue, serine (S), the 155th residue, and glutamine (Q), the 317th residue. The serine residue forms the active center of the rat IAP. Rat IAPs are coordinated by three metal ions, one Mg^2+^ and two Zn^2+^, which play an important role in the enzyme activity [30]. However, the physiological and biochemical implications and the affinity of the common IIAEK-binding amino acid sequence (GFYLFVEGGR) for IAPs obtained from rat small intestinal mucosal proteins and Caco-2 cell-derived intestinal lipid raft fractions will require further investigation in the future.

AP, including IAP, occurs in two forms: as a glycosylphosphatidylinositol (GPI)-anchored protein [31,32] and as a secreted protein (non GPI-anchored protein) [31]. The GPI-anchored protein is a receptor that covalently binds to glycosylphosphatidylinositol (GPI), a saturated phospholipid. GPI-anchored proteins reside in a signaling microdomain called a lipid raft, which is present in the plasma membrane [32]. For example, the GPI-anchored protein CD59, a dimer, forms a stable tetramer following extracellular ligand stimulation, leading to the induction of associated intracellular signaling (IP_3_-Ca^2+^) [33]. In mice, the GPI-anchored protein CD55 regulates osteoclast function through receptor activator-associated Rac signaling of NF-κβ ligands [34]. Only a few studies have reported on GPI-anchored IAP-mediated signaling. However, as GPI-anchored proteins such as CD59 [33] and CD55 [34] regulate physiological functions via intracellular signaling, the existence of an analogous novel signaling system mediated by GPI-anchored IAP is highly probable. Interestingly, our results verify the existence of such a novel signaling pathway associated with the interaction of IIAEK and IAP, which is expressed at high levels in the small intestine.

Since our experimental results suggest that IIAEK upregulates IAP expression and downregulates ABCA1 expression via GPI-anchored IAP, we focused on the association between IAP and ABCA1. The transcription factors that bind to the IAP promoter include HNF-4α [35], KLF5 [36], the TR-RXRα heterodimer [37], ZBP-89 [38], and Cdx-1 [39], and the transcription factors binding to the ABCA1 promoter include LXR/RXR (transcriptional activation), TRβ/RXR (transcriptional suppression), and SREBP-2 [21]. The existence of a common transcription factor that binds to both the IAP and ABCA1 gene promoter is highly likely. However, further investigations are required to determine whether IIAEK affects such a common transcription factor. Our present results suggest that IIAEK improves cholesterol metabolism through an increase of IAP mRNA levels and a decrease of ABCA1 mRNA levels in Caco-2 cells.

The current results revealed that IIAEK significantly increased IAP mRNA, protein levels, and AP activity, while significantly decreasing ABCA1 mRNA and protein levels (Figure 1B–D and Figure 5A–J). Comparison of the IIAEK-mediated increase of IAP mRNA, protein levels, and AP activity to a positive control with known effects on these parameters or a dose–response assessment will be performed in our future study. The physiological functions of IAP are associated with the regulation of homeostasis in the small intestinal epithelial environment, including fatty acid absorption in the small intestine [40], detoxification of inflammatory mediators by dephosphorylation [41,42], pH regulation of the duodenal surface [43], inhibition of LPS-induced inflammatory responses [24], and regulation of the tight junction-associated proteins at the genetic level [44]. Interestingly, IAP is closely associated with lipid metabolism. Previous studies have suggested that the increased absorption of fatty acids and serum triglycerides in the small intestine in IAP KO mice is related to the fatty acid transporter CD36, which is present in the vicinity of AKP6, one of the murine IAPs [45]. Further studies of the underlying molecular mechanisms are necessary to understand the relationship between IAP and lipid metabolism.

In our previous study, we reported a significant increase in serum HDL cholesterol levels and a significant decrease in serum LDL+VLDL in rats administered IIAEK orally compared those receiving casein tryptic hydrolysate (CTH) [3]. A significant reduction in total serum cholesterol levels was also observed in rats administered IIAEK orally compared to rats receiving beta-sitosterol [3]. Our current experimental findings revealed that the addition of 2 mM IIAEK significantly increased IAP gene expression, protein levels, and AP activity in Caco-2 cells incubated with or without cholesterol micelles. Based on previous reports as well as the current experimental findings, we propose that IAP is involved in cholesterol metabolism, which is intricately related to lifestyle diseases such as atherosclerosis and myocardial infarction.

Interestingly, IAP KO mice show symptoms of metabolic disorders including elevated plasma cholesterol levels, elevated serum triglyceride levels, and abnormal glucose tolerance [46]. IAP has been shown to prevent aging, contribute to a prolonged lifespan, target the intestinal barrier, and to improve the disruption of intestinal permeability in mice [47]. Furthermore, the oral administration of IAP significantly decreased serum triglyceride and LDL cholesterol levels and significantly increased serum HDL cholesterol levels in mice [46,47]. However, no reports exist on the relationship between GPI-anchored IAPs and cholesterol metabolism. Most reports of IAPs associated with lipid metabolism have only discussed the functions of IAPs secreted as non-GPI-anchored IAPs [31], while few research groups have focused on the functions of GPI-anchored IAPs [31,32]. Although a close association between IAP and lipid metabolism has been previously proposed, the relationship between IAP and cholesterol metabolism has been rarely addressed. Our findings reveal a novel signaling pathway involving IIAEK, a ligand that interacts with GPI-anchored IAPs to induce intracellular signaling, thereby up-regulating IAP gene expression and protein levels while downregulating ABCA1 expression and protein levels, which is accompanied by the inhibition of cholesterol absorption. We emphasized the role of IAP as a novel target molecule in the regulation of cholesterol metabolism.

## 5. Conclusions

In conclusion, IIAEK interacts with GPI-anchored IAP, which is highly expressed in small intestinal epithelial cells and improves cholesterol metabolism through the upregulation of IAP and the downregulation of ABCA1 through intracellular signaling (Figure 6). Our findings contribute to the elucidation of the physiological response to oligopeptides at the cell membrane and the subsequent intracellular signal transduction pathways associated with cholesterol metabolism.

## Figures and Tables

**Figure 1 nutrients-12-02859-f001:**
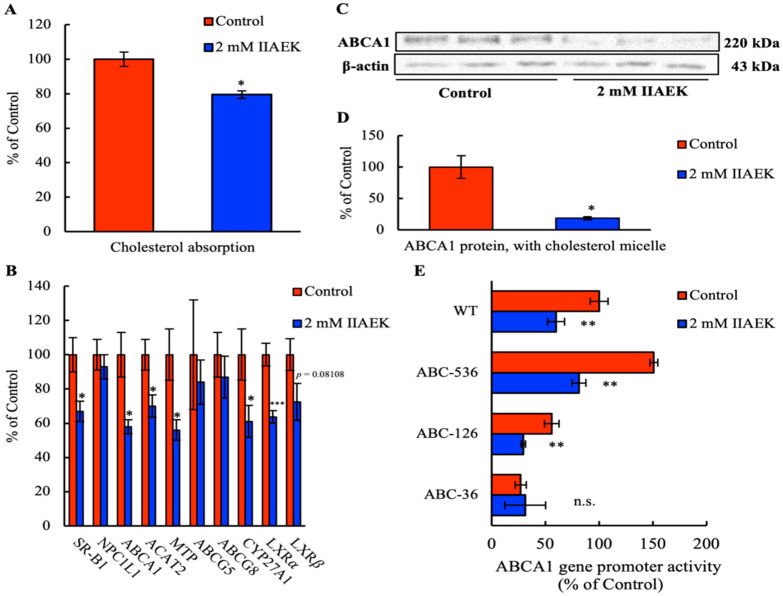
Effect of IIAEK (Ile-Ile-Ala-Glu-Lys, lactostatin) containing cholesterol micelle on cholesterol metabolism in Caco-2 cells. (**A**) Effect of IIAEK containing cholesterol micelle on cholesterol absorption rate in Caco-2 cells. (**B**) Effect of IIAEK containing cholesterol micelle on cholesterol metabolism-associated gene expression in Caco-2 cells. Values are represented as means ± standard errors, represented by vertical bars (*n* = 6 per group). (**C**) Effect of IIAEK containing cholesterol micelle on ATP-binding cassette transporter A1 (ABCA1) protein level in Caco-2 cells by Western blot. (**D**) ABCA1 protein level was quantified with ImageJ and normalized to the level of β-actin. Values are represented as mean ± standard error, represented by vertical bars (*n* = 3 per group). (**E**) Effect of IIAEK containing cholesterol micelle on ABCA1 gene promoter activity in Caco-2 cells. Caco-2 cells were transfected with the human pGL3-ABCA1-Luc plasmid (WT: −928 to +107 bp) or the pGK3-ABCA1-Luc plasmid deletion mutants (ABC–536: −536 to +107 bp, ABC–126: −126 to +107 bp, ABC–36: −36 to +107 bp), using the pPGK β-galactosidase plasmid as an internal control. Data are presented as luciferase activity normalized to β-galactosidase activity. Values are represented as means ± standard error, represented by vertical bars (*n* = 5–9 per group). Asterisks indicate the difference from the control (* *p* < 0.05, ** *p* < 0.01, ****p* < 0.001), as determined by Student’s *t*-test.

**Figure 2 nutrients-12-02859-f002:**
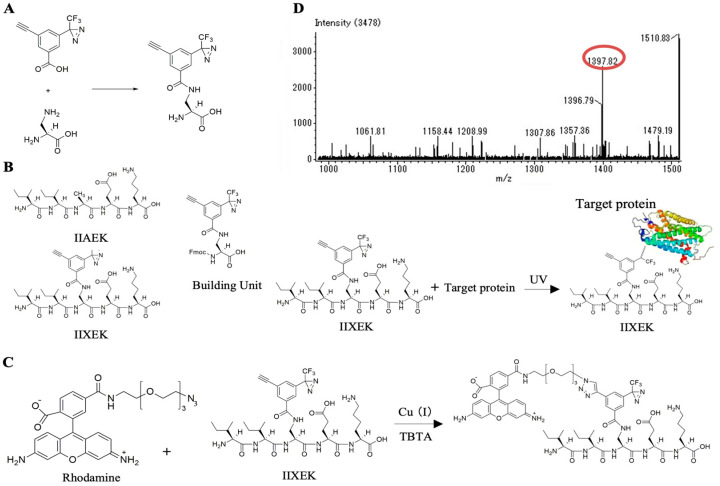
Chemical synthesis of the molecular probe, IIXEK. (**A**) Amide bond by 3-ethynyl-5-[3-(trifluoromethyl)-3H-diazirin-3-yl] benzoic acid and (S)-2,3-diaminopropionic acid. (**B**) Structure of IIXEK. IIXEK possesses chemically synthesized 3-ethynyl-5-[3-(trifluoromethyl)-3H-diazirin-3-yl] benzoic acid. This probe captures the target proteins interacting with IIAEK. (**C**) Click reaction, which occurs by introduction of rhodamine into IIXEK in the presence of copper as catalyst (Huisgen cycloaddition reaction). (**D**) Mass spectrometry of Cu-Catalyzed Azide (Azide-fluor 488) Alkyne (IIXEK) cycloaddition.

**Figure 3 nutrients-12-02859-f003:**
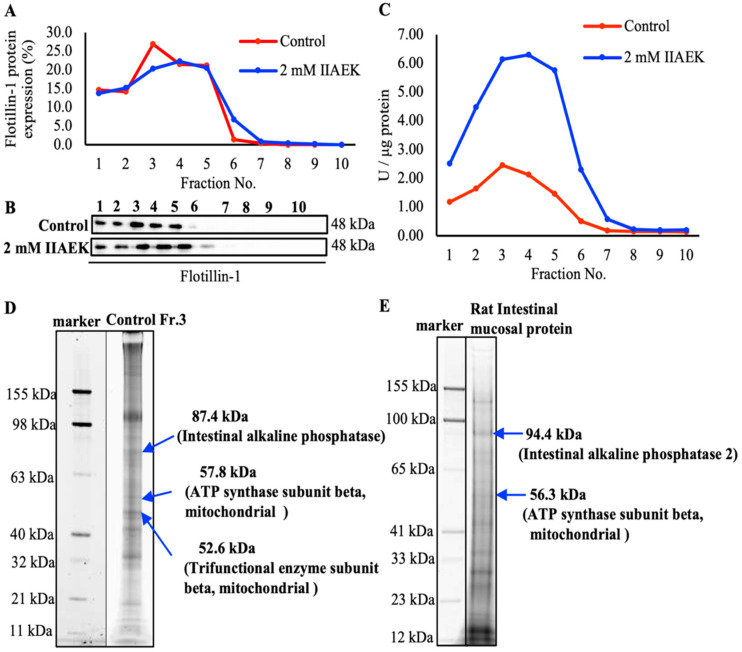
Extraction of intestinal lipid raft fractions from Caco-2 cells and photoaffinity labeling by IIXEK for the lipid raft fraction and rat intestinal mucosal protein. (**A**,**B**) Protein level of the lipid raft marker flotilin-1 in each fraction. (**C**) Alkaline phosphatase (AP) activity in each fraction. (**D**) Photoaffinity labeling of intestinal lipid raft proteins by IIXEK. The intestinal lipid raft fraction (Fr. 3) containing 250 μM IIXEK was irradiated with UV for 30 min for photoaffinity labeling. Subsequently, the fraction containing the fluorescent substance, rhodamine, which was added by a click reaction, was separated on SDS-PAGE. (**E**) Photoaffinity labeling of rat intestinal mucosal protein fractions by IIXEK. We collected the small intestinal mucosal proteins of 5-week-old male Wistar rats. The membrane protein fractions, which were derived from rat small intestinal mucosa, were collected with EB2B of TM-PEK and made into rat small intestinal mucosal fractions. Photoaffinity labeling was performed in the same way as described in Figure 3D.

**Figure 4 nutrients-12-02859-f004:**
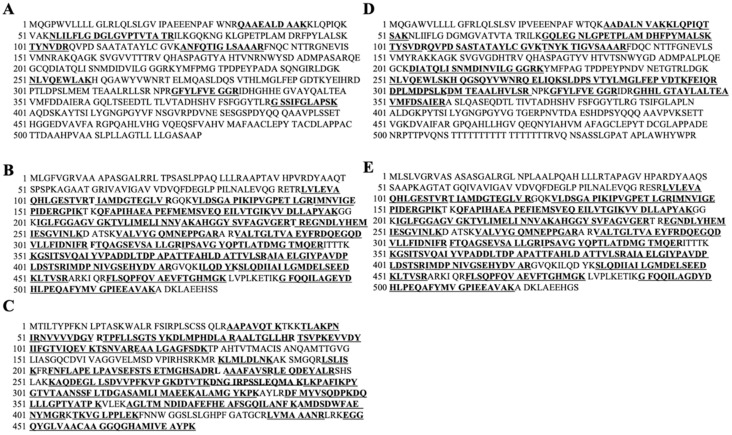
IIXEK-binding amino acid sequences of proteins identified by nano LC-MS/MS analyses. (**A**) Human IAP (Uniprot accession No. P09923). (**B**) Human ATP synthase subunit beta, mitochondrial (Uniprot accession No. P06576). (**C**) Human Trifunctional enzyme subunit beta, mitochondrial (Uniprot accession No. P55084). (**D**) Rat IAP-2 (Uniprot accession No. P51740). (**E**) Rat ATP synthase subunit beta, mitochondrial (Uniprot accession No. P10719). Matched peptide sequences are bold-faced and underlined.

**Figure 5 nutrients-12-02859-f005:**
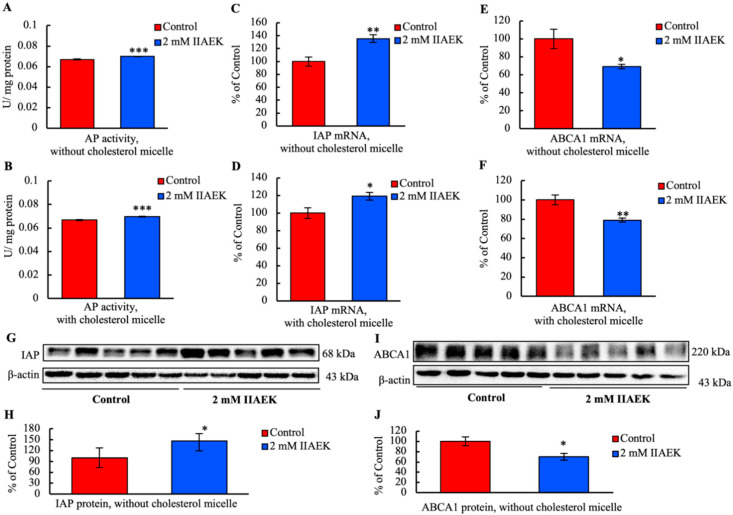
Effects of IIAEK on the expression of intestinal alkaline phosphatase (IAP) and the cholesterol metabolism-associated gene, ABCA1. Caco-2 cells were cultured in six-well Transwell plates. The cells were treated with serum-free medium containing 2 mM IIAEK or vehicle for 24 h. They were incubated with or without cholesterol micelle. (**A**,**B**) The total proteins were collected from the cells, and alkaline phosphatase (AP) activity was measured. Values are represented as means ± standard error represented by vertical bars (*n* = 6 per group). (**C**–**F**) Total RNA was collected from the cells. The IAP and ABCA1 mRNA levels were measured by real-time PCR and normalized to the mRNA expression level of 18S ribosomal RNA. Values are represented as means ± standard error represented by vertical bars (*n* = 6 per group). (**G**,**I**) The cell lysate was collected from the cell, separated on SDS-PAGE, and analyzed by Western blot analysis. (**H**,**J**) The IAP and ABCA1 protein levels were quantified with ImageJ and normalized to the level of β-actin. Values are represented as means ± standard error represented by vertical bars (*n* = 5 per group). Asterisks indicate the difference from the control (* *p* < 0.05, ** *p* < 0.01, *** *p* < 0.001) as determined by Student’s *t*-test.

**Figure 6 nutrients-12-02859-f006:**
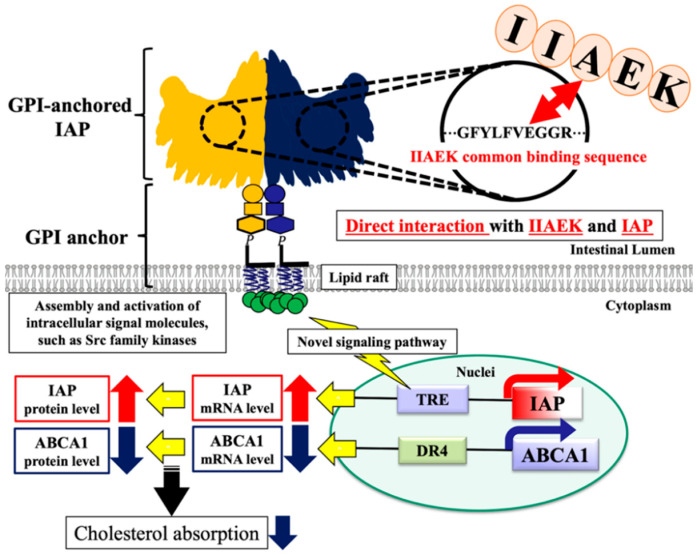
IIAEK–IAP interaction hypothesis. IIAEK interacts with IAP and improves cholesterol metabolism by the specific activation of IAP and downregulation of intestinal ABCA1.

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
