# Peer review of "IIAEK Targets Intestinal Alkaline Phosphatase (IAP) to Improve Cholesterol Metabolism with a Specific Activation of IAP and Downregulation of ABCA1"

_nutrients, 2020, doi:10.3390/nu12092859_

Round 1

Reviewer 1 Report

The paper explores the potential mechanism by which the peptide IIAEK lowers cholesterol. The paper fits the scope of the journal. This work is a continuation of the research done by this group which discovered the peptide.

A major review of the English language is needed. It is hard to understand the manuscript as is.

In Figure 2C, rhodamine should be labeled. Consider labeling the last structure as IIXEK plus rhodamine.

Under the discussion section (lines 447-460), it is not clear why didn’t the authors consider the molecular weight of rhodamine if this is the molecule responsible for making the complex fluorescent? Wouldn’t the weight of 86.82387 be of the IAP + IIXEK+rhodamine? The same question applies to the 93.82387 kDa complex

Author Response

[Our Reply for Reviewer 1 Report]

We are grateful for the comments from Reviewers on our manuscript, as these were very insightful and helped us to improve our manuscript.

We corrected many sentences with a red color in revised manuscript.

[Reviewer 1 Report]

The paper explores the potential mechanism by which the peptide IIAEK lowers cholesterol. The paper fits the scope of the journal. This work is a continuation of the research done by this group which discovered the peptide.

1.A major review of the English language is needed. It is hard to understand the manuscript as is.

(Our reply 1) Thank you very much for your valuable comments.

We have extensively reviewed English language of our manuscript.

2.In Figure 2C, rhodamine should be labeled. Consider labeling the last structure as IIXEK plus rhodamine.

(Our reply 2) According to reviewer’s suggestion, rhodamine was labeled in Figure 2C.

3.Under the discussion section (lines 447-460), it is not clear why didn’t the authors consider the molecular weight of rhodamine if this is the molecule responsible for making the complex fluorescent? Wouldn’t the weight of 86.82387 be of the IAP + IIXEK+rhodamine? The same question applies to the 93.82387 kDa complex

(Our replay 3) The molecular weight of a rhodamine in this experiment is 574.58 Da. As reviewer’s suggestion, taking into account the molecular weight of a rhodamine, we revised the 86.82347 kDa to 87.39847 kDa and the 93.82387 kDa to 94.39847 kDa in revised manuscript (P.12, L433-467). By these changes in the molecular weights, the molecular weights of IAPs also faintly changed in Figure 3 in revised manuscript (Figure 3 and P.8, L332-340).

Reviewer 2 Report

The authors study the effect of a penta-peptide IIAEK from bovine milk on the molecular mechanisms of cholesterol metabolism. The authors have performed a number of assays to investigate the mechanism. But there are several issues that need to be addressed.

  1. In lines 238-242, and in other places, values are reported without specifying units or descriptions. It is not clear if these are raw values, the authors should mention what these values represent for each experiment.
  2. In figure 3D, photoaffinity labeling by IIXEK shows a number of bands on the SDS PAGE gel. How did the authors conclude that IIXEK only interacted with 86.8 kDa band (IAP). To show specificity, the authors could have used a control peptide, to show that this peptide did not label IAP.
  3. The authors do not report or explain the results from MS analysis used to find the amino acid sequence interacting with IIXEK.
  4. The method used for measuring IAP activity is not explained properly. In line 164-173, after incubation at 37C how was the activity measured and converted to U/mg of protein. In figure 4A, the increase of IAP activity by IIAEK is minuscule, yet the authors say that there is a highly significant increase of IAP activity (p<0.001). Comparison to a positive control with known increase of IAP activity or a dose response curve is needed to resolve this discrepancy.
  5. Similarly, western blot results in figure 4G and 4H are not convincing without a positive control or dose response.
  6. The significance of figure 5 is not clear. Addition of IAP siRNA does not decrease ABCA1 mRNA levels. How is this related to IIAEK? Was the effect of IAP siRNA tested in the presence of IIAEK? The authors do not provide clear interpretation for this data.
  7. In line 451, the authors state that  they could detect 0.82387 kDa difference by SDS-PAGE, which to me does not seem possible. Besides, no controls were run without the peptide bound to IAP to show that the peptide binding shifts the molecular weight by 0.82387 kDa.

Author Response

[Our Reply for Reviewer 2 Report]

We are grateful for the comments from Reviewers on our manuscript, as these were very insightful and helped us to improve our manuscript.

We corrected many sentences with a red color in revised manuscript.

[Reviewer 2 Report]

The authors study the effect of a penta-peptide IIAEK from bovine milk on the molecular mechanisms of cholesterol metabolism. The authors have performed a number of assays to investigate the mechanism. But there are several issues that need to be addressed.

1.In lines 238-242, and in other places, values are reported without specifying units or descriptions. It is not clear if these are raw values, the authors should mention what these values represent for each experiment.

(Our reply 1) Thank you very much for your valuable comments.

As our data expressions in the text were very confused in original manuscript as reviewer’s suggestion, we deleted the data expressions (values are reported without specifying units or descriptions) in the Results (P.5, L223 – P.11, L406) in original manuscript.

2.In figure 3D, photoaffinity labeling by IIXEK shows a number of bands on the SDS PAGE gel. How did the authors conclude that IIXEK only interacted with 86.8 kDa band (IAP). To show specificity, the authors could have used a control peptide, to show that this peptide did not label IAP.

(Our reply 2) Before we respond your suggestion 2, we have faintly changed the molecular weight of IAP in Figure 3 according to the reviewer 1’s suggestion 3. As reviewer 2’s suggestion, there are a number of bands on SDS-PAGE gel in the photoaffinity labeling by IIAEK using the lipid raft fractions of Caco-2 cells (Figure 3D) or rat intestinal mucosal protein (Figure 3E). At first, we concluded that 87.4 kDa band was identified as a human IAP from MALDI-TOF-MS data (sequences coverage,14.77%, Figure 4A). Matched peptide sequences are bold-faced and underlined. Also, we concluded that 94.4 kDa band was identified as a rat IAP-2 from MALDI-TOF-MS data (sequences coverage, 37.02%, Figure 4B). Matched peptide sequences are bold-faced and underlined. Moreover, 87.4 kDa band or 94.4 kDa band contained the amino acid sequence of IAP including GFYLFVEGGR (common IIAEK-binding amino acid sequence)by MS analysis (Figure 4A, B). Of note, since the binding specificity of IIXEK for IAP during photoaffinity labeling is not particularly high, as revealed by SDS-PAGE (Figure 3D, E), we also investigated the relationships between IAP and IIAEK in the lipid raft fractions from Caco-2 cells. Surprisingly, IAP activity was drastically increased by IIAEK in the lipid raft fractions, which is of great importance with regard to cellular signal transduction, including cholesterol metabolism (Figure 3C). Finally, IIAEK significantly increased both the IAP mRNA level and IAP activity in Caco-2 cells. Taken together, we have concluded that an important interaction occurs between IAP and IIAEK. In order to show the specificity of the interaction between IIAEK and IAP, we need to use a control peptide in a future study and confirm that this peptide will not label IAP.

These explanations are involved in revised manuscript (Figure 3, Figure 4, P.8, L.333-345, P.12, L.442- L.474)

3.The authors do not report or explain the results from MS analysis used to find the amino acid sequence interacting with IIXEK.

(Our reply 3) We have addressed this question in our reply 2.

4.The method used for measuring IAP activity is not explained properly. In line 164-173, after incubation at 37°C how was the activity measured and converted to U/mg of protein. In figure 4A, the increase of IAP activity by IIAEK is minuscule, yet the authors say that there is a highly significant increase of IAP activity (p<0.001). Comparison to a positive control with known increase of IAP activity or a dose response curve is needed to resolve this discrepancy.

(Our reply 4) After 30 min of incubation at 37°C. the absorbance at 405 nm was measured, and IAP activity was calculated as μmol/min using a calibration curve for various concentrations of p-nitrophenol. μmol/min was defined as U. U was converted to U/mg protein in order to normalize IAP activity to the protein concentration, which was determined using a commercially available kit (BioRad, protein assay; BioRad).

Moreover, as reviewer’s suggestion, comparison between the IIAEK-mediated increase of IAP protein levels and activity to a positive control with known effects on these parameters or a dose-response assessment will be performed in our future study.

These explanations are involved in revised manuscript (P.4, L.166-178, P.13, L.515-519)

5.Similarly, western blot results in figure 4G and 4H are not convincing without a positive control or dose response.

(Our reply 5) As reviewer’s suggestion, comparison between the IIAEK-mediated increase of IAP protein levels and activity to a positive control with known effects on these parameters or a dose-response assessment will be performed in our future study.

These explanations are involved in revised manuscript (P.13, L.515-519)

6.The significance of figure 5 is not clear. Addition of IAP siRNA does not decrease ABCA1 mRNA levels. How is this related to IIAEK? Was the effect of IAP siRNA tested in the presence of IIAEK? The authors do not provide clear interpretation for this data.

(Our reply 6) It has been reported that IAP contributes to the improvement of lipid metabolism. Our present results suggest that IIAEK improves cholesterol metabolism by an increase of IAP mRNA level and a decrease of ABCA1 mRNA level in Caco-2 cells. Thus, our data suggest that changes in the mRNA levels of IAP and ABCA1 are inversely related. Thus, we have predicted that a decrease of IAP mRNA level will not change or increase ABCA1 mRNA level by IAP siRNA treatment in Caco-2 cells. In fact, ABCA1 mRNA level has unchanged by IAP siRNA treatment as predicted (Figure 5). As reviewer’s suggestion, our future studies should further assess and measure the effects of IAP siRNA treatment in the presence of IIAEK.

These explanations are involved in revised manuscript (P.13, L.508-514)

7.In line 451, the authors state that they could detect 0.82387 kDa difference by SDS-PAGE, which to me does not seem possible. Besides, no controls were run without the peptide bound to IAP to show that the peptide binding shifts the molecular weight by 0.82387 kDa.

(Our reply 7) As reviewer’s suggestion, it is hard to detect 0.82387 kDa difference by SDS-PAGE. The molecular weight of IIXEK (0.82387 kDa) was calculated from the structural formula of IIXEK in the text. In figure 2D, click reaction product formation of IIXEK and rhodamine has been confirmed. As reviewer’s suggestion, setting the control run without IIXEK to show that IIXEK binding shifts the molecular weight by 0.82387 kDa will be considered in the future.

These explanations are involved in revised manuscript (P.12, L.442-445, L.456-458)

Round 2

Reviewer 1 Report

Minor editing is still required (e.g line 447: " Therefore, it was observed that one IIXEK molecules..." The word molecule should not be plural.

Author Response

[Our Reply for Reviewer 1 Report]

We are grateful for the comments from Reviewers on our manuscript, as these were very insightful and helped us to improve our manuscript.

We corrected Figures and some sentences with a red color in revised manuscript.

[Reviewer 1 Report]

1.Minor editing is still required (e.g line 447: " Therefore, it was observed that one IIXEK molecules..." The word molecule should not be plural.

(Our reply 1) According to reviewer’s suggestion, we corrected the sentence in revised manuscript(line 490).

Reviewer 2 Report

The authors did not address the concerns satisfactorily. There are still several issues with results and interpretations

  1. The specificity of photoaffinity labeling by IIXEK is still a concern. Though the authors say that IAP activity was drastically increased by IIAEK in the lipid raft fractions, they measure bulk phosphatase activity in the fraction and not IAP specific phosphatase activity. So, this does not address the specificity issue
  2. The authors state that changes in the mRNA levels of IAP and ABCA1 are inversely related. So, a decrease in IAP mRNA should increase ABCA1 mRNA, but they detected no change, which is counter to their explanation
  3. The MALDI-TOF data and methods were not provided. The authors just provide sequences they detected for the 87.4 kDa band. Its not clear what are the other bands in SDS-PAGE that are photo-labeled
  4. Controls in other experiments are also missing as noted before which was not addressed

Author Response

[Our Reply for Reviewer 2 Report]

We are grateful for the comments from Reviewers on our manuscript, as these were very insightful and helped us to improve our manuscript.

We corrected Figures and some sentences with a red color in revised manuscript.

[Reviewer 2 Report]

The authors did not address the concerns satisfactorily. There are still several issues with results and interpretations

1.The specificity of photoaffinity labeling by IIXEK is still a concern. Though the authors say that IAP activity was drastically increased by IIAEK in the lipid raft fractions, they measure bulk phosphatase activity in the fraction and not IAP specific phosphatase activity. So, this does not address the specificity issue.

(Our reply 1) As reviewer’s suggestion, our IAP activity in the lipid raft fractions is not IAP specific phosphatase activity. Thus, we modified the sentences about the specificity of photoaffinity labeling by IIXEK in revised manuscript as follows.

Of note, a number of bands were observed on the SDS-PAGE gel following photoaffinity labeling by IIXEK using lipid raft fractions from Caco-2 cells (Figure 3D) or rat intestinal mucosal protein (Figure 3E). At first, we concluded that 87.4 kDa band represented human IAP identified from nano LC-MS/MS data (sequences coverage, 14.77%, Figure 4A). We also concluded that the 94.4 kDa band was identified as rat IAP revealed by nano LC-MS/MS data (sequences coverage, 37.02%, Figure 4D). Moreover, both the 87.4 kDa band and 94.4 kDa band contained the amino acid sequence of IAP, including the GFYLFVEGGR (common IIAEK-binding amino acid sequence) revealed by MS analysis (Figure 4A, D). Since the binding specificity of IIXEK for IAP during photoaffinity labeling is not particularly high, as revealed by SDS-PAGE (Figure 3D, E), we also investigated the relationships between IAP and IIAEK in the lipid raft fractions from Caco-2 cells. It has been reported that IAP activity consists of IAP specific phosphatase activity and tissue-non-specific alkaline phosphatase (TNAP) activity in Caco-2 cells [15]. Also, it has been reported that about 75% of IAP activity is IAP specific phosphatase activity and about 25 % of IAP activity is TNAP in Caco-2 cells [15]. As it is well known that IAP is a typical intestinal lipid raft marker as well as flotillin-1, we measured IAP specific phosphatase activity to confirm the formation of the lipid raft and found that IAP specific phosphatase activity was drastically increased by IIAEK in the lipid raft fractions, which is of great importance with regard to cellular signal transduction, including cholesterol metabolism (Figure 3C). Furthermore, IAP protein level was significantly increased by IIAEK treatment compared to control in Caco-2 cells (Figure 5G). Finally, IIAEK significantly increased both the IAP mRNA level and IAP activity in Caco-2 cells. Taken together, we have concluded that an important interaction occurs between IAP and IIAEK. In order to show the specificity of the interaction between IIAEK and IAP, we need to use a control peptide in a future study and confirm that this peptide will not label IAP.

These explanations are involved in revised manuscript (Figure 3, 4, line 204-217, 335-342, 501-524)

2.The authors state that changes in the mRNA levels of IAP and ABCA1 are inversely related. So, a decrease in IAP mRNA should increase ABCA1 mRNA, but they detected no change, which is counter to their explanation.

(Our reply 2) As reviewer’s suggestion, we revised the manuscript because our statement that changes in the mRNA levels of IAP and ABCA1 are inversely related are exaggerated. Thus, we modified the sentences about the siRNA experiment as follows.

Further, the current observations suggest that changes in the mRNA levels of IAP and ABCA1 are inversely related. Thus, we have hypothesized that the changes in the mRNA levels of IAP and ABCA1 are inversely related. It was observed that the knockdown of IAP by IAP siRNA was canceled the hypothetical inverse relationships between IAP and ABCA1 (Figure 6). Our future studies should further assess and measure the effects of IAP siRNA treatment in the presence of IIAEK.

These explanations are involved in revised manuscript (line 560-564).

3.The MALDI-TOF data and methods were not provided. The authors just provide sequences they detected for the 87.4 kDa band. Its not clear what are the other bands in SDS-PAGE that are photo-labeled.

(Our reply 2) According to reviewer’s suggestion, we modified the sentences about the nano LC-MS/MS experiment in revised manuscript as follows.

[Materilas and Methods]

Subsequently, this mixture was subjected to 4-12% SDS-PAGE for fluorescence scanning. The BenchMark™ Fluorescent Protein Standard was used as the molecular weight marker. The photoaffinity labeled protein was analyzed by nano LC-MS/MS (Q Exactive Plus, Thermo Scientific). Mascot search engine (www.matrixscience.com) was used for database search within UniprotKB/Swiss-Prot database to match with the parent proteins.

These explanations are involved in revised manuscript (line 204-217)

(Our reply 2) As reviewer’s suggestion, we show what are the other bands in SDS-PAGE that are photo-labeled in the revised manuscript as follows.

[Figure, Results]

In addition, we concluded that the 94.4 kDa band was identified as rat IAP-2 revealed by nano LC-MS/MS data (sequences coverage, 37.02 %, Figure 4D). Matched peptide sequences are bold-faced and underlined. Interestingly, comparison of the IIXEK-binding amino acid sequences of IAP in intestinal lipid raft fraction of Caco-2 cells (IAP) (Figure 4A) and rat small intestinal mucosal protein fractions (IAP-2) (Figure 4D) by MS analysis revealed that GFYLFVEGGR was the common IIAEK-binding amino acid sequence, and their position within human IAP (Uniprot accession No. P09923) from N to C terminal:324-333 and rat IAP-2 (Uniprot accession No. P51740) from 324-333 (Figure 4A, D). In the lipid raft fraction from Caco-2 cells, we identified 57.8 kDa band as ATP synthase subunit beta, mitochondrial (coverage, 74.67%, Uniprot accession No. P06576, Figure 4B) and 52.6 kDa band as Trifunctional enzyme subunit beta, mitochondrial (coverage, 71.52%, Uniprot accession No. P55084, Figure 4C) from nano LC-MS/MS data sequences. Matched peptide sequences are bold-faced and underlined. In rat mucosal protein, we identified 56.3 kDa band as ATP synthase subunit beta, mitochondrial (coverage, 74.67%, Uniprot accession No. P10719, Figure 4E) from nano LC-MS/MS data sequences. Matched peptide sequences are bold-faced and underlined.

These explanations and data are involved in revised manuscript (Figure 3 and 4, line 340-354)

4.Controls in other experiments are also missing as noted before which was not addressed.

(Our reply 4) According to reviewer’s suggestion, we modified the sentences about all experiment as follows.

The current results revealed that IIAEK significantly increased IAP mRNA, protein levels, and enzyme activity, while significantly decreasing ABCA1 mRNA and protein levels (Figure 1B-D, 5A-J). Comparison of the IIAEK-mediated increase of IAP mRNA, protein levels and enzyme activity to a positive control with known effects on these parameters or a dose-response assessment will be performed in our future study.

These explanations are involved in revised manuscript (line 565-569)
